



# Sea ice transport and replenishment across and within the Canadian Arctic Archipelago: 2016-2022

Stephen E.L. Howell[1], David G. Babb[2], Jack C. Landy[3], Isolde A. Glissenaar[4], Katlin McNeil[5], Benoit Montpetit[1], and Mike Brady[1]

[1]Climate Research Division, Environment and Climate Change Canada, Toronto, Canada
[2]Centre for Earth Observation Science, University of Manitoba, Winnipeg, Canada
[3]Department of Physics and Technology, The Arctic University of Norway, Tromsø, Norway
[4]Bristol Glaciology Centre, School of Geographical Sciences, Bristol, United Kingdom
[5]Department of Physics, University of Toronto, Toronto, Ontario, Canada

*Correspondence to*: S.E.L. Howell (stephen.howell@ec.gc.ca)

**Abstract.** The Canadian Arctic Archipelago (CAA) serves as both a source and sink for sea ice from the Arctic Ocean, while also exporting sea ice into Baffin Bay. We use observations from Sentinel-1, RADARSAT-2, the RADARSAT Constellation Mission (RCM), and CryoSat-2 together with the Canadian Ice Service ice charts to quantify sea ice transport and replenishment across and within the CAA from 2016 to 2022. We also provide the first estimates of the ice area and volume flux within the CAA from the Queen Elizabeth Islands to the Parry Channel which spans the central region of the Northwest Passage shipping route. Results indicate that the CAA primarily exports ice to the Arctic Ocean and Baffin Bay with an average annual (October to September) ice area flux of $134\pm72\times10^3$ km$^2$ and a volume flux of $40\pm74$ km$^3$. The CAA contributes a larger area but smaller volume of ice downstream to the North Atlantic than what is delivered via Nares Strait. The average annual ice area flux from the Queen Elizabeth Islands to the Parry Channel was $27\pm10\times10^3$ km$^2$ and the volume flux was $34\pm12$ km$^3$, with a majority occurring through Byam Martin Channel which is directly above the central region of Northwest Passage. Over our study period, annual multi-year ice (MYI) replenishment within the CAA was resilient with an average of $16\pm49\times10^3$ km$^2$ imported from the Arctic Ocean, and an average of $56\pm36\times10^3$ km$^2$ of first-year ice (FYI) retained following the melt season. The considerable ice flux to the Parry Channel together with sustained MYI replenishment emphasizes the continued risk that sea ice poses to practical utilization of key shipping routes in the CAA, including the Northwest Passage.

## 1 Introduction

The Canadian Arctic Archipelago (CAA) is a collection of islands on the North American continental shelf that is bounded by the Arctic Ocean to the west and Baffin Bay to the east (Figure 1). Therefore, the channels, straits, and inlets of the CAA are important pathways where both freshwater and human goods are transported. In terms of freshwater, the CAA is a primary pathway for transporting freshwater from the Arctic Ocean to the North Atlantic via Baffin Bay (Steele et al., 1996;



Prinsenberg and Hamilton, 2005; Jones et al., 2003; Rudels, 2011; Zhang et al., 2021), where it has implications for large-scale meridional overturning circulation (Kuhlbrodt et al., 2007). In terms of transporting goods, the Northwest Passage bisects the CAA and provides a shorter path than the Northern Sea Route for connecting the Atlantic and Pacific Oceans.

Under a warming environment, the ice pack within the CAA has declined in area and produced a longer melt season (Howell et al., 2009) that has led to an increase in shipping activity within the CAA since the 1990's (Pizzolato et al., 2014; Dawson et al., 2018). Interest in the practical usage of the Northwest Passage continues to grow as climate models project its sea ice cover to decline (e.g. Smith and Stephenson, 2011; Mudryk et al., 2021) together with the political uncertainty of utilizing the Northern Sea Route along the Russian Arctic coast (Vylegzhanin et al., 2020; Li and Lynch, 2023). Therefore, it is

crucial to understand the current patterns of sea ice transport across and within the CAA in order to adapt to changes in shipping activity (Dawson et al., 2020).

Sea ice within the CAA is a mix of seasonal first-year ice (FYI) and perennial multi-year ice (MYI). The ice cover is typically landfast from November to July (Canadian Ice Service, 2021), during which time the ice is immobile and separated

from the mobile pack ice beyond the CAA by stable ice arches that routinely form across the straits that bound the CAA (Figure 1). When the ice in the CAA melts during spring, areas of open water form and the ice becomes mobile, creating a narrow window for ice dynamics to take place. Due to this brief window for ice dynamics to occur, the CAA is responding differently to climate change. For example, Melling (2022) recently reported similar ice thickness values within the northern CAA, or the Queen Elizabeth Islands (QEI), between the 1970s and 40 years later in 2009-2010 and suggested that the

dynamic processes that facilitate thick ice north of the CAA have been less impacted by climate change. Moreover, Glissenaar et al. (2023) found no change in sea ice thickness along Parry Channel from January to April between 1996 and 2020, compared to dramatic thinning in both the Beaufort Sea and Baffin Bay regions on either side of the CAA. There is also evidence that the sea ice area flux from the Arctic Ocean into the QEI has increased (Howell and Brady, 2019) and the dynamic processes that transport thick MYI from the Arctic Ocean into the CAA have exhibited no signs of stopping

(Howell et al., 2023a). Regardless of when the Arctic Ocean becomes seasonally free of sea ice (Kim et al., 2023; Topál and Ding, 2023) and loses its MYI pack (Babb et al., 2023), the remaining reservoir of MYI located north of the CAA (i.e. the Last Ice Area) is still expected to flow southward into the CAA and maintain a thick ice pack.

High spatial resolution SAR satellite imagery is ideally suited for monitoring sea ice dynamics in the CAA because of its

numerous narrow channels, straits, and inlets that are more difficult to resolve in coarser sea ice derived satellite products. The availability of SAR imagery across the CAA since the launch of RADARSAT-1 in 1995 and RADARSAT-2 in 2007 has allowed for the ice area flux between the Arctic Ocean and the CAA to be quantified (Kwok, 2006; Howell et al., 2013; Howell and Brady, 2019). Unfortunately, image availability of RADARSAT-1 and RADARSAT-2 was not spatially and temporally consistent over the entire CAA and as a result, a complete picture of sea ice dynamics of the CAA over the entire

annual cycle was not possible with SAR. Indeed, Agnew et al. (2008) quantified sea ice dynamic processes (i.e. sea ice



motion and ice area flux) across the CAA from September 2002 to June 2007 using enhanced resolution AMSR-E at 89 GHz satellite observations but atmospheric interference at 89 GHz prevented estimates during the summer months, when the ice pack is mobile. The relatively recent availability of Sentinel-1 followed by the RADARSAT Constellation Mission (RCM) has essentially transformed the availability of high spatiotemporal resolution SAR imagery across the entire CAA (Howell et

al., 2022). Moreover, there have been recent breakthrough's using CryoSat-2 that allow Arctic sea ice thickness to be estimated during the summer as well as winter months (Landy et al., 2022) and also within the prominently landfast regions of the CAA (Glissenaar et al., 2023). These recent developments present a new opportunity for improving our understanding of sea ice dynamics within and across the CAA on an annual basis.

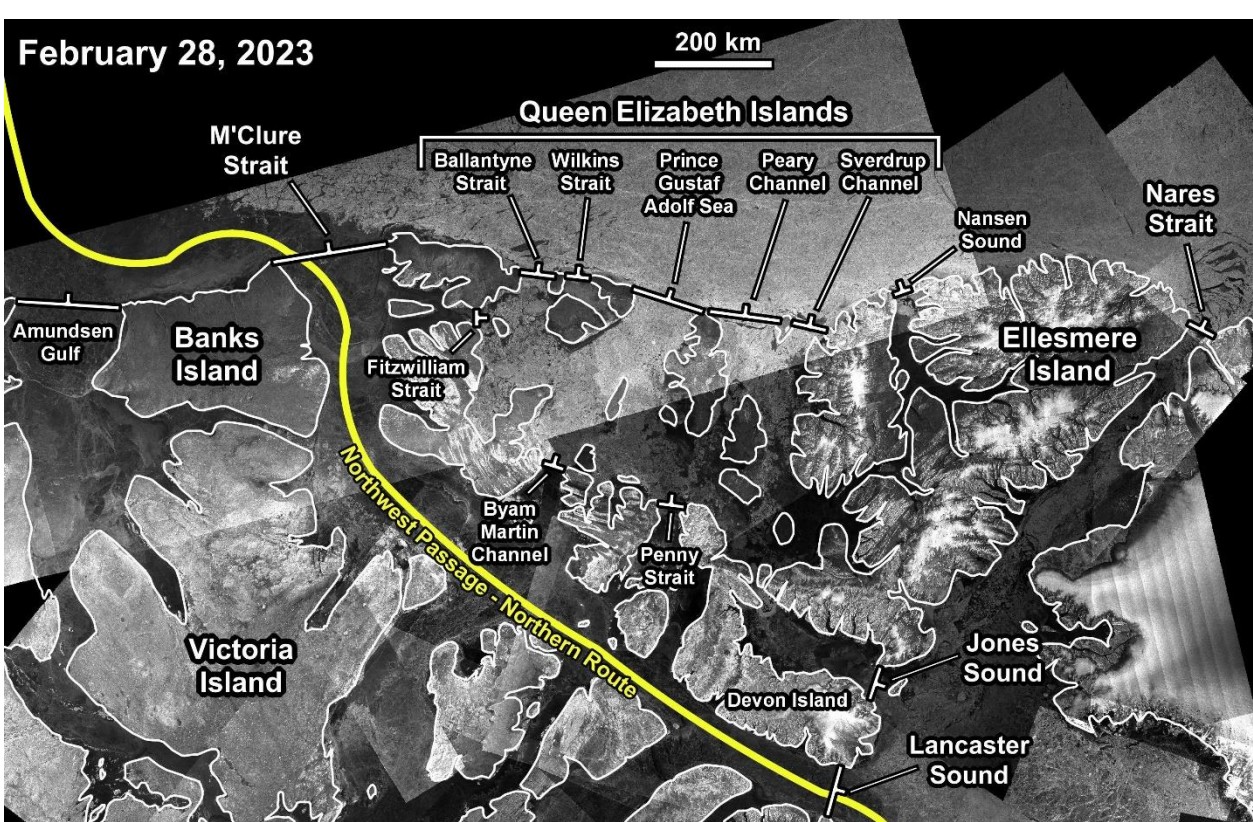


**Figure 1: Map of the Canadian Arctic Archipelago with the location of the of all the sea ice flux gates used in this study. Background is RADARSAT Constellation Mission (RCM) imagery on February 28, 2023. (RCM © Government of Canada).**

Here, we use Sentintel-1, RADARSAT-2, RCM, and CryoSat-2 together with the Canadian Ice Service ice charts to quantify

the sea ice area flux from October 2016 to September 2022 (6-years) and the volume fluxes from October 2016 to September 2020 (4-years) across the CAA and discuss their annual and interannual variability. To our knowledge, this is the first study to provide coincident ice area and volume flux estimates across all of the boundaries of the CAA, in particular the eastern



boundaries to Baffin Bay, which have received considerably less attention. We also consider sea ice area and volume transport within the CAA, specifically between the QEI and Parry Channel, which is a key part of the Northwest Passage

where ice transport is widely known to occur (Melling, 2002; Howell et al., 2009; Howell et al., 2023a) but has never been quantified. Finally, we use our flux estimates to provide a more robust estimate of MYI replenishment within the CAA on an annual basis.

## 2 Data

The primary data used in this analysis was synthetic aperture radar (SAR) imagery from RADARSAT-2 (2016-2020),

Sentintel-1 (2016-2021), and the RCM (2020-2022) at HH polarization. RADARSAT-2 and RCM imagery is available online at Natural Resources Canada's Earth Observation Data Management System (https://www.eodms-sgdot.nrcan-rncan.gc.ca). Sentinel-1 imagery is available at the Copernicus Open Access Hub (https://scihub.copernicus.eu/dhus/#/home). All images were resampled to a spatial resolution of 200 m. The average temporal sample window over all gates in the CAA was ~1 day and the record spans from October 2016 to September 2022.


Sea ice thickness estimates for the outer flux gates were acquired from the CryoSat-2 radar altimeter from October 2016 to July 2021 using a combination of data from Landy et al., (2020) for the 'cold' season (October to April) and Dawson et al., (2022) for the summer period (May to September). A bias correction based on radar model simulations (Landy et al., 2022) is applied to the summer radar freeboards. The entire time series of radar freeboards is then converted to a continuous pan-

Arctic record of sea ice thickness using snow loading information from SnowModel-LG (Stroeve et al., 2020) and assuming constant ice-type dependent densities for sea ice (Landy et al., 2022). All CryoSat-2 ice thickness data are available from https://data.bas.ac.uk/full-record.php?id=GB/NERC/BAS/PDC/01613. No thickness measurements are available from August 2021 to September 2022.

Sea ice thickness estimates for the inner flux gates within the CAA were obtained from the ice thickness proxy record developed by Glissenaar et al. (2023), which is available from https://doi.org/10.5281/zenodo.7644084. This proxy sea ice thickness dataset uses CryoSat-2 observations of sea ice thickness in the open seas of the Canadian Arctic to train a random forest regression model to estimate sea ice thickness from information in the Canadian Ice Service ice charts (Tivy et al., 2011) within the channels of the CAA. Unlike direct CryoSat-2 observations, this proxy sea ice thickness dataset is available

in all channels in the CAA for November to April. The uncertainty of the sea ice thickness values range from 30 to 50 cm.

Additional supporting data used in this analysis include weekly total, MYI, and second-year ice (SYI) concentration from the Canadian Ice Service digital ice charts (Tivy et al., 2011).



## 3 Methods

The sea ice area flux for all outer gates of the CAA and the interior gates of the CAA was estimated from October 2016 to
September 2022 (Figure 1). The outer flux gates facing the Arctic Ocean are Amundsen Gulf, M'Clure Strait and the QEI
with the QEI gates collectively comprised of Ballantyne Strait, Wilkins Strait, Prince Gustaf Adolf Sea, Peary Channel,
Sverdrup Channel, and Nansen Sound. Amundsen Gulf has an aperture of 169 km, M'Clure Strait has an aperture of 183 km,
and the total aperture of all QEI gates is 405 km. The outer flux gates facing Baffin Bay are Jones Sound and Lancaster

Sound with apertures of 58 km and 83 km, respectively. The inner flux gates of the CAA were chosen as the three gates that
connect the QEI to the southern half of the CAA and the northwest passage; the three gates are Fitzwilliam Strait (34 km
aperture), Byam Martin Channel (35 km aperture), and Penny Strait (49 km aperture).

Our approach to estimate the sea ice area flux from sequential pairs of SAR imagery is robust and based on previous work

(e.g. Kwok, 2006; Agnew et al., 2008; Howell et al., 2013). For each SAR image pair, sea ice motion was estimated using
the Environment and Climate Change Canada Automated Sea Ice Tracking System (ECCC-ASITS; Howell et al., 2022) that
is based on the Komarov and Barber (2013) feature tracking algorithm. Sea ice motion estimates are then interpolated to a 30
km buffer region at each gate and then sampled at 5 km intervals. The sea ice area flux ($F_A$) was then calculated using the
following equation:

$$F_A = \sum c_i u_i \, \Delta x \qquad (1)$$

where, $\Delta x$ is the spacing along each gate (i.e., 5 km), $u_i$ is the ice motion normal to the flux gate at the $i^{th}$ location and $c_i$ is the
sea ice concentration determined from the Canadian Ice Service ice charts. For the outer gates, positive flux values represent
CAA ice import (i.e. Arctic Ocean or Baffin Bay ice import into the CAA) and negative flux values represent CAA ice
export (i.e. ice export into the Arctic Ocean or Baffin Bay). For the inner gates, positive flux values represent southward

transport of sea ice from the QEI and negative values represent northward transport into the QEI. For all gates, the sea ice
area flux values were summed over each month from October 2016 to September 2022.

The uncertainty ($\sigma_{FA}$) in $F_A$ can be estimated following Kwok and Rothrock (1999) by assuming errors in sea ice motion are
additive, uncorrelated, and normally distributed using the following equation:

$$\sigma_{FA} = \frac{\sigma_u L}{\sqrt{N_s}} \qquad (2)$$

where, $\sigma_u$ is the error in SAR derived ice motion, $L$ is the width of the gate, and $N_s$ is the number of samples across the gate.
The value of $\sigma_u$ has been found to range from 0.43-3.43 km depending on the ice conditions for the region and the time of
year (Lindsay and Stern, 2003; Komarov and Barber, 2013; Howell et al., 2022). The upper bound considers all vectors at
the pan-Arctic scale together with no stringent conditions for buoy comparison and therefore is likely too high given slower

ice movement and higher concentrations within the CAA hence, we constrain the upper bound to 3 km. $\sigma_u$ is likely at the
lower range for M'Clure Strait, QEI, Fitzwilliam Strait, Byam Martin Channel, and Penny Strait since sea ice is typically



found in high concentration all year around (CIS, 2021). At Amundsen Gulf, Jones Sound, and Lancaster Sound gates, $\sigma_u$ could potentially approach the upper bound in some months because these gates are seasonally ice-free during summer. The ice area flux uncertainty on a monthly basis ($\sigma_T$) can subsequently be estimated using the following equation:

$$\sigma_T = \sigma_{FA}\sqrt{N_D} \tag{3}$$

where, $N_D$ is the number of observations per month (~30). Table 1 shows $\sigma_T$ from solving equations (2) and (3) with a range of $\sigma_u$.

The sea ice volume flux of the CAA's outer and inner gates from October 2016 to July 2021 was determined from the product of the monthly ice area flux and the monthly average CryoSat-2 sea ice thickness within the 30 km buffer around each gate. Sea ice thickness was not available coincident with the sea ice area flux at the inner CAA flux gates therefore, we used the linear trend in April to November sea ice thickness values to approximate thickness values from May to October. We estimate the uncertainty ($\sigma_{FV}$) in the ice volume flux following Kwok and Rothrock (1998) using:

$$\sigma_{FV} = \sqrt{(F_A\sigma_h)^2 + (h\sigma_T)^2} \tag{4}$$

where, $h$ is the ice thickness and $\sigma_h$ is the uncertainty in thickness. $\sigma_h$ for the outer gates is taken from Landy et al. (2022). $\sigma_h$ for the inner gates has been found to range from 30 to 50 cm (Glissenaar et al. (2023) and accordingly, we have taken the average (40 cm) as the inner gate uncertainty. Table 2 summarizes $\sigma_{FV}$ from solving equation (3) for the upper and lower bounds of $\sigma_u$.

**Table 1: The uncertainty in monthly sea ice area flux ($\sigma_T$) for the upper and lower range in the error in SAR derived ice motion ($\sigma_e$) for each gate.**

| Gate Name | Area Flux Uncertainty ($\sigma_T$, km²) | |
| --- | --- | --- |
| | Lower | Upper |
| Amundsen Gulf | 68 | 478 |
| M'Clure Strait | 71 | 497 |
| Queen Elizabeth Islands | 106 | 739 |
| Jones Sound | 40 | 280 |
| Lancaster Sound | 48 | 335 |
| Fitzwilliam Strait | 31 | 214 |
| Byam Martin Channel | 31 | 217 |
| Penny Strait | 37 | 257 |



**Table 2: The uncertainty in monthly sea ice volume flux ($\sigma_{FV}$) for the upper and lower range in the error in SAR derived ice motion ($\sigma_e$) for each gate.**

| Gate Name | Volume Flux Uncertainty ($\sigma_{FV}$, km$^3$) | |
|---|---|---|
| | Lower | Upper |
| Amundsen Gulf | 2.3 | 2.4 |
| M'Clure Strait | 1.7 | 2.0 |
| Queen Elizabeth Islands | 1.9 | 3.0 |
| Jones Sound | 0.1 | 0.2 |
| Lancaster Sound | 2.1 | 2.1 |
| Fitzwilliam Strait | 0.1 | 0.4 |
| Byam Martin Channel | 0.7 | 0.9 |
| Penny Strait | 0.3 | 0.5 |

## 4 Results and Discussion

### 4.1 CAA monthly and annual area and volume flux

The time series of monthly sea ice area and volume flux across the outer gates of the CAA from October 2016 to September 2022 together with their average climatological seasonal cycle (October to September) is shown in Figure 2. The average monthly ice area flux (± standard deviation) was -11±25x10$^3$ km$^2$ and ranged from -70x10$^3$ km$^2$ to 55x10$^3$ km$^2$ (Figure 2a) while the average monthly ice volume flux (± standard deviation) was -5±25 km$^3$ and ranged from -65 km$^3$ to 55 km$^3$ (Figure 2c). For both ice area and volume flux, the monthly variability was substantial but in general the CAA imported ice area from July to September and exported ice area from October to June (Figure 2b). Ice volume was imported from June to November and exported from December to May (Figure 2d).

The annual (October to September) ice area flux for the entire CAA and its individual outer gates are shown in Figure 3. Over the 6-year period, the average annual ice area flux was -134±72x10$^3$ km$^2$ and ranged from -274x10$^3$ km$^2$ in 2019 to -37x10$^3$ km$^2$ in 2020. On an annual basis, the CAA exported more ice to the Arctic Ocean and Baffin Bay than it received, though ice is typically only imported via the small apertures of the QEI. On average, the annual ice area flux across the boundaries of the CAA is ~15% of the long-term average Fram Strait ice area flux (880x10$^3$ km$^2$; Smedsrud et al. 2017). Compared to recent seasonal ice area flux estimates for Nares Strait by Howell et al. (2023b) for 2017 to 2021 (95x10$^3$ km$^2$) the average seasonal ice area flux for the CAA is ~141% of Nares Strait.

The annual (October to September) ice volume flux for all of the CAA together with its individual outer exchange gates are shown in Figure 4. The average volume flux was -40±74 km$^3$, indicating a net export of ice out of the CAA over the 4-year period. Considerable interannual variability in the CAA ice volume flux was apparent with minimal export in 2017 (-5 km$^3$), strong export in 2019 (-145 km$^3$) and strong import in 2020 (55 km$^3$). The CAA's annual ice volume flux across the outer boundaries of the CAA corresponds to ~4% of Fram Strait using the average ice volume from 2010 to 2018 (990 km$^3$)



estimated by Sumata et al. (2022) and ~23% of Nares Strait using average annual ice volume flux in Nares Strait from 2017
to 2020 (177 km$^3$) estimated by Howell et al. (2023b).

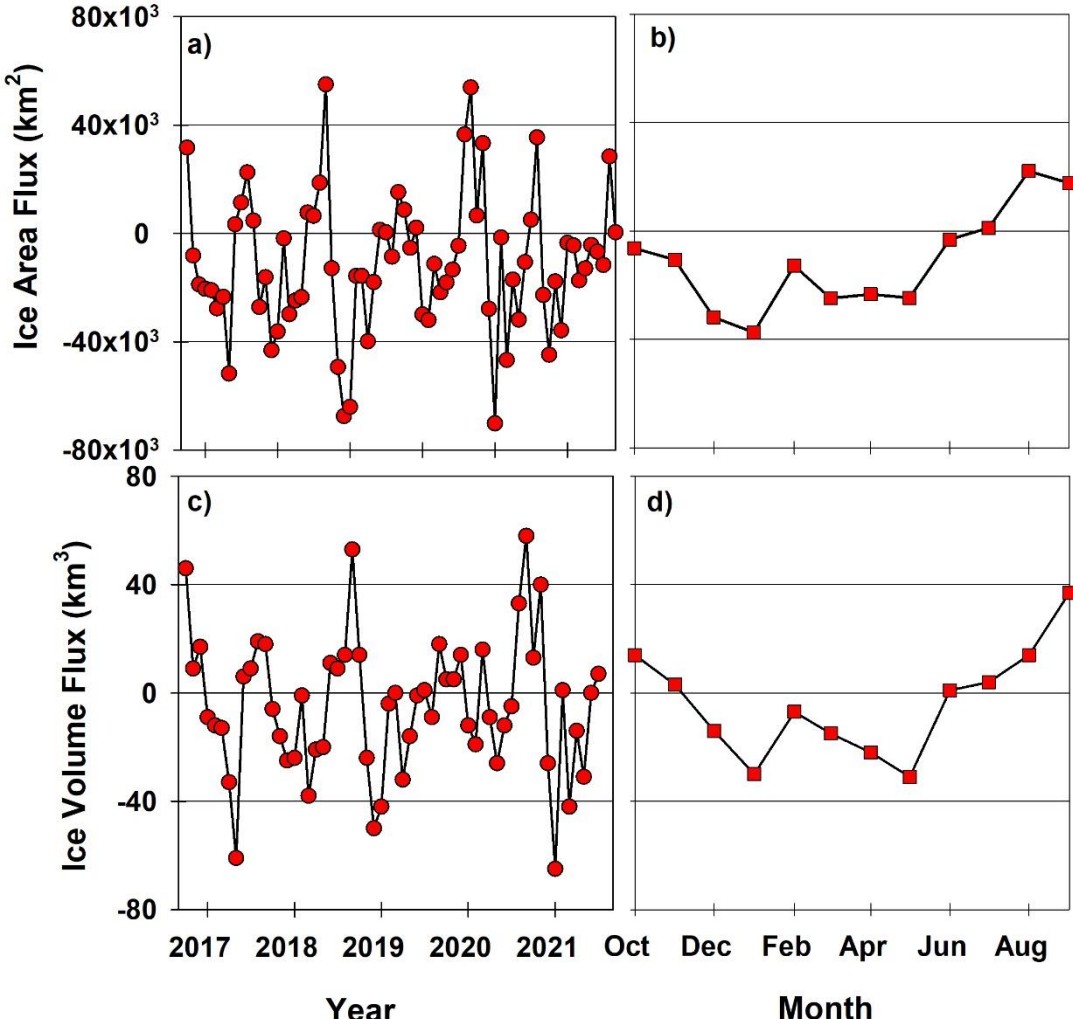

**Figure 2: (a) time series of the monthly ice area flux from October 2016 to September 2022, (b) the average monthly ice area flux, (c) time series of the ice volume flux from October 2016 to September 2022, and (d) the average monthly ice volume flux. Positive**
**flux values indicate ice import to the Arctic Ocean or Baffin Bay and negative values indicate ice export from the Arctic Ocean or Baffin Bay.**




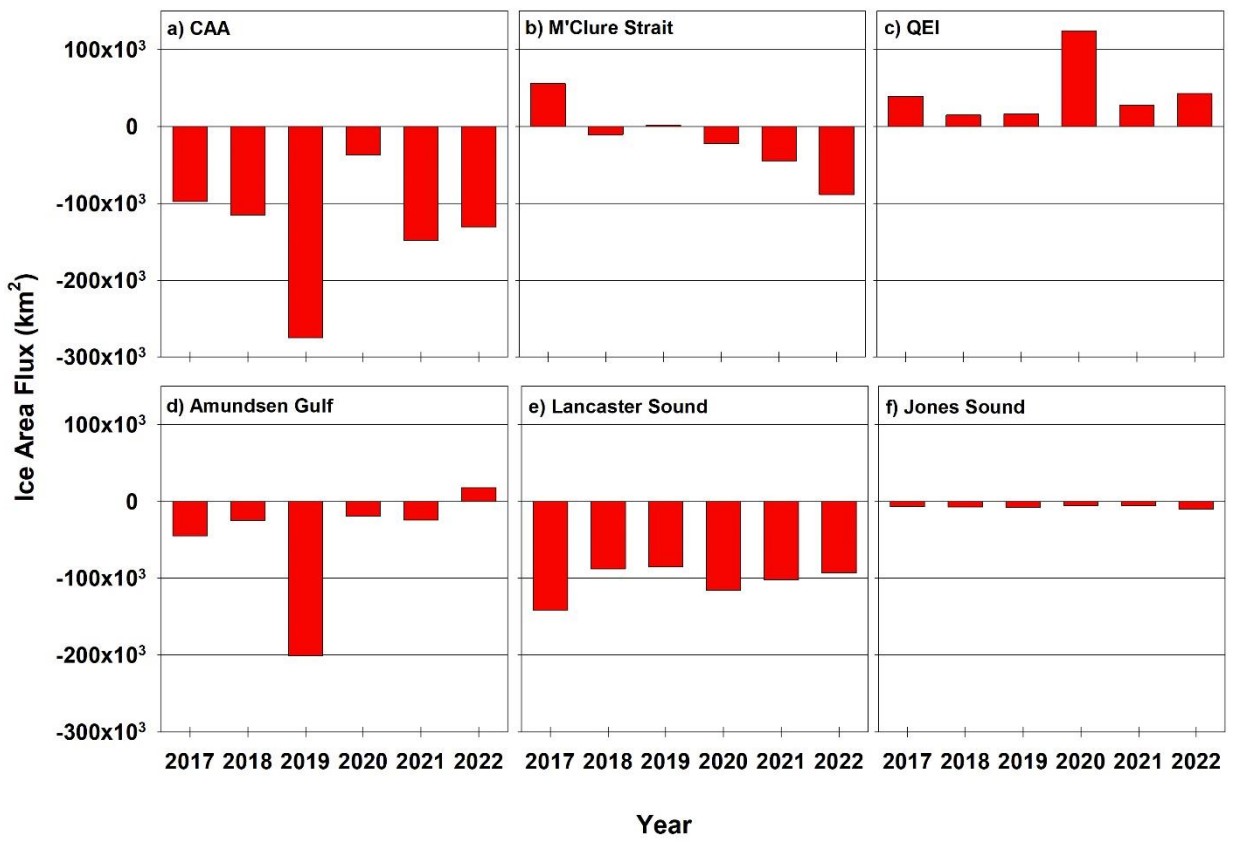

**Figure 3. Annual (October to September) ice area ice flux (a) the Canadian Arctic Archipelago (CAA), (b) M'Clure Strait, (c) Queen Elizabeth Islands (QEI), (d) Amundsen Gulf, (e) Lancaster Sound, and (f) Jones Sound for 2017 to 2022. Positive flux values indicate ice import from the Arctic Ocean or Baffin Bay and negative values indicate ice export to the Arctic Ocean or**
**Baffin Bay.**



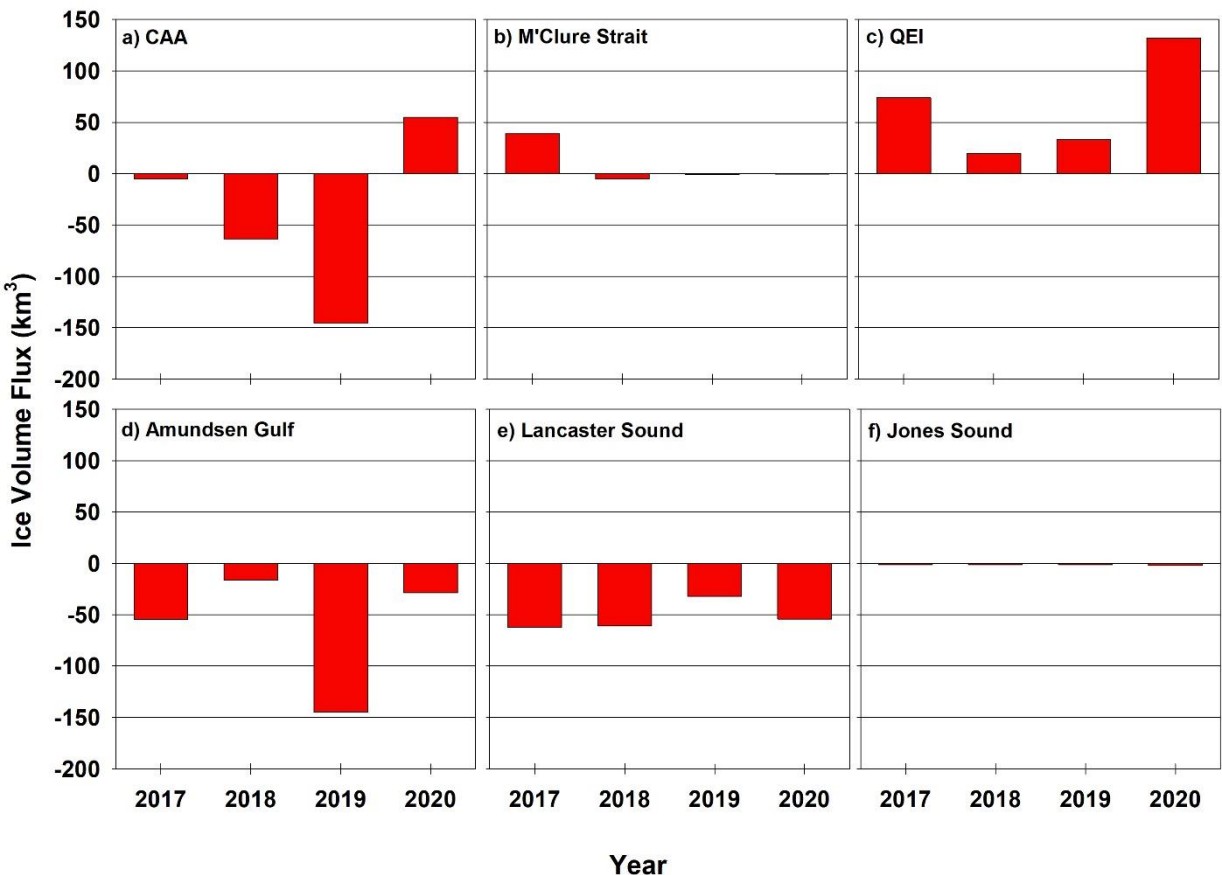


**Figure 4: Annual (October to September) ice volume flux (a) the Canadian Arctic Archipelago (CAA), (b) M'Clure Strait, (c) Queen Elizabeth Islands (QEI), (d) Amundsen Gulf, (e) Lancaster Sound, and (f) Jones Sound for 2017 to 2022. Positive flux values indicate ice import from the Arctic Ocean or Baffin Bay and negative values indicate ice export to the Arctic Ocean or Baffin Bay.**


## 4.2 Ice flux comparison between the Arctic Ocean and Baffin Bay

For the gates facing the Arctic Ocean, Amundsen Gulf was the primary export gate while for the gates facing Baffin Bay, Lancaster Sound was the primary export gate (Figure 3; Figure 4). Sea ice was only imported via the gates facing the Arctic Ocean and this primarily occurred through the QEI, though sea ice was imported once through M'Clure Strait and

Amundsen Gulf. The majority of sea ice area flux was towards Baffin Bay with a 6-year annual average of $111\pm19\text{x}10^3$ km$^2$ or 83% of the total sea ice flux, meaning that the remaining $23\pm84\text{x}10^3$ km$^2$ or 17% was exported into the Arctic Ocean (Figure 3). With respect to ice volume flux, the 4-year annual average indicates that $12\pm81$ km$^3$ was imported from the Arctic Ocean and $54\pm12$ km$^3$ was exported to Baffin Bay (Figure 4). In terms of the ice flux delivered downstream to the



North Atlantic via Baffin Bay on an annual basis, the CAA was a larger contributor than Nares Strait for ice area but not for
ice volume, because the majority of sea ice being exported from the CAA was FYI compared to MYI from Nares Strait.

Comparing the primary export gate for the Arctic Ocean and Baffin Bay indicates more ice export and less interannual
variability for Lancaster Sound compared to Amundsen Gulf. The annual 6-year average ice area flux for the Amundsen
Gulf was $-49\pm70$ x$10^3$ km$^2$ and ranged from $-200$x$10^3$ km$^2$ in 2019 to 18x$10^3$ km$^2$ in 2022 (Figure 3). The annual 6-year
average at Lancaster Sound was $-104\pm20$ x$10^3$ km$^2$ and ranged from $-142$x$10^3$ km$^2$ in 2017 to $-85$x$10^3$ km$^2$ in 2019 (Figure
3). In terms of the annual ice volume flux, Amundsen Gulf volume flux was larger than Lancaster Sound with 4-year annual
averages of $-61\pm50$ km$^3$ and $-52\pm12$ km$^3$, respectively (Figure 4).  It is also worth emphasizing that the CAA's large net
annual ice export in 2019 was driven by the large anomalous export at Amundsen Gulf (Figure 4). The time series of
cumulative ice area flux at Amundsen Gulf during the anomalous year of 2019 together with a more typical year of 2021 is
shown in Figure 5. Ice area export during winter 2019 was particularly pronounced and despite a prolonged period of no flux
during summer 2019 (Figure 5). Export occurred almost continuously from October 2018 through May 2019, after which
there was very little export during summer 2019 when the area is typically ice-free (Figure 5). For comparison, ice export
during 2020-2021 was characterized by ice import during November 2020, followed by episodic export through to May 2021
and limited flux during summer. The Amundsen Gulf region is part of a larger flaw-lead polynya system that forms
throughout the southeastern Beaufort Sea (Carmack and MacDonald, 2002; Barber and Hanesiak, 2004; Galley et al., 2008).
We suggest increased ice area export during 2019 was primarily due to the fact that the ice arch that typically forms across
Amundsen Gulf never formed (not shown), allowing the ice pack to remain mobile and more responsive to synoptic winds
that flushed ice out of Amundsen Gulf into the Beaufort Sea.

The primary pathway for sea ice import into the CAA was the QEI, which had a 6-year average annual ice area flux of
$44\pm37$x$10^3$ km$^2$ that ranged from 15x$10^3$ km$^2$ in 2018 to 125x$10^3$ km$^2$ in 2020 (Figure 3).The 4-year average annual volume
flux was $65\pm44$ km$^3$ and ranged from 20 km$^3$ in 2018 to 132 km$^3$ in 2020 (Figure 4). The processes responsible for the large
ice flux into the QEI in 2020 was previously discussed by Howell et al. (2023b) who reported that the amount even exceeded
the largest reported annual ice flux at Nares Strait and was approximately 10% of the average sea ice volume export through
Fram Strait from 2010-2018 (Sumata et al., 2022). The oldest and thickest sea ice in the world lies just north of the QEI
(Kwok, 2018; Landy et al., 2022) and this thick ice certainly plays a considerable role in modulating the annual ice area and
volume flux for the CAA via the QEI. For example, the Arctic Ocean ice area flux into the QEI in 2020 was not sufficient to
balance ice area export from the other regions but the large Arctic Ocean ice volume flux in 2020 compensated for the
volume of ice exported at all the other passageways and actually led to a net import of sea ice volume into the CAA (Figure
4a). 2017 was similar such that the 74 km$^3$ of Arctic Ocean ice imported through the QEI contributed to a very small annual
volume flux from the CAA (-5 km$^3$). Overall, the CAA appears to be a strong ice area exporter and a lower ice volume
exporter when appreciable ice from the Arctic Ocean is imported to the QEI.



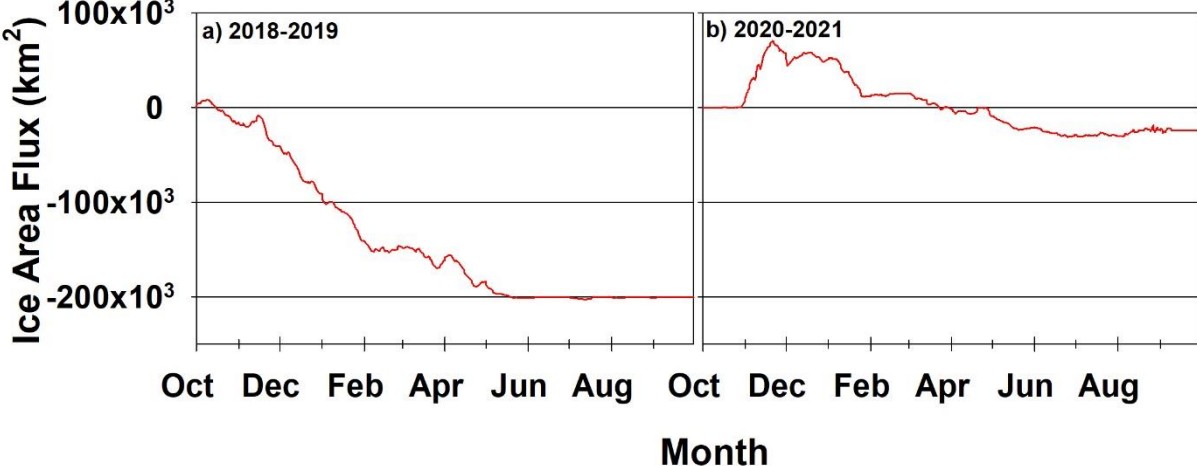

**Figure 5: Time series of the Amundsen Gulf cumulative ice area flux for (a) 2018-2019 and (b) 2020-2021. Positive flux values indicate ice import from Arctic Ocean and negative values indicate ice export to the Arctic Ocean.**

### 4.3 Ice area and volume transport from the QEI to the Parry Channel

The spatial distribution of MYI within the CAA at the end of the melt season for 2017 to 2022 is shown in Figure 6. It is evident from Figure 6 that MYI can be observed to be flowing southward from QEI to the Parry Channel via the CAA's internal passageways of Fitzwilliam Strait, Byam Martin Channel, and Penny Strait. While this process of southward advection or flushing from the QEI has long been known to occur (i.e. Melling, 2002; Alt et al., 2006; Howell et al., 2009) it has never been quantified.

Figure 7 shows the monthly time series of the sea ice area flux at the CAA's internal passageways. It is clear that ice flux only occurs during summer when the ice pack is mobile and that the majority of ice exiting the QEI passes through Byam Martin Channel. The average monthly ice area flux through Byam Martin Channel is $2\pm3\times10^3$ km$^2$ but importantly this can range from $1\times10^3$ km$^2$ to $20\times10^3$ km$^2$. The monthly mean ice area fluxes through Penny Strait and Fitzwilliam Strait are $0.5\pm1\times10^3$ km$^2$ and $-0.03\pm0.4\times10^3$ km$^2$, respectively. These average ice area flux values are relatively small because theeregions remain landfast for much of the year resulting in minimal sea ice motion outside of August and September when the majority of the flux occurred.

Considering all three passageways, the average annual ice area flux from the QEI over the 6-year period was $27\pm10\times10^3$ km$^2$ and ranged from $9\times10^3$ km$^2$ in 2019 to $40\times10^3$ km$^2$ in 2020 with MYI representing about ~39% of the total ice area flux (Figure 8). The average annual volume flux was $34\pm12$ km$^3$ and ranged from 13 km$^3$ in 2019 to 49 km$^3$ in 2020 (Figure 8).



The ice exiting out of these passageways subsequently flows into Parry Channel, where it represents a considerable risk to ships transiting the Northwest Passage. Moreover, Haas and Howell (2015) reported mean ice thickness values in Byam Martin Channel to be 3.84 m and recent analysis by Melling (2022) suggests that dynamic thickening of old multi-year ice in the Arctic Ocean immediately north of the CAA has not declined in such a way that the ice imported into the QEI has

significantly thinned since the 1970s. Ultimately, this suggests that thick ice continues to be advected from the Arctic Ocean into the QEI and subsequently transported southwards to the Northwest Passage where it poses a risk to ships navigating the area. The risk of encountering thick sea ice in the Northwest Passage has not disappeared, in spite of significant overall thinning of the Arctic ice cover.

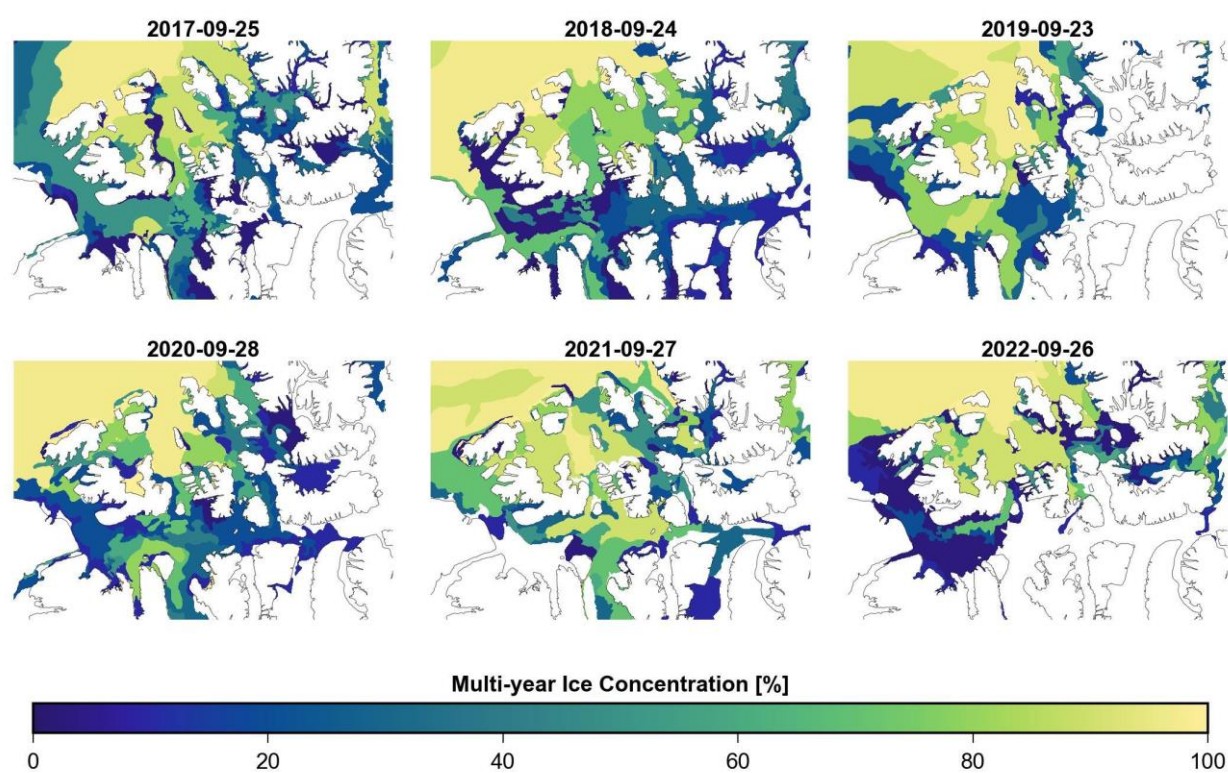


**Figure 6: Spatial distribution of multi-year ice (MYI) concentration on the last week of September in the Canadian Arctic Archipelago for 2017 to 2022.**






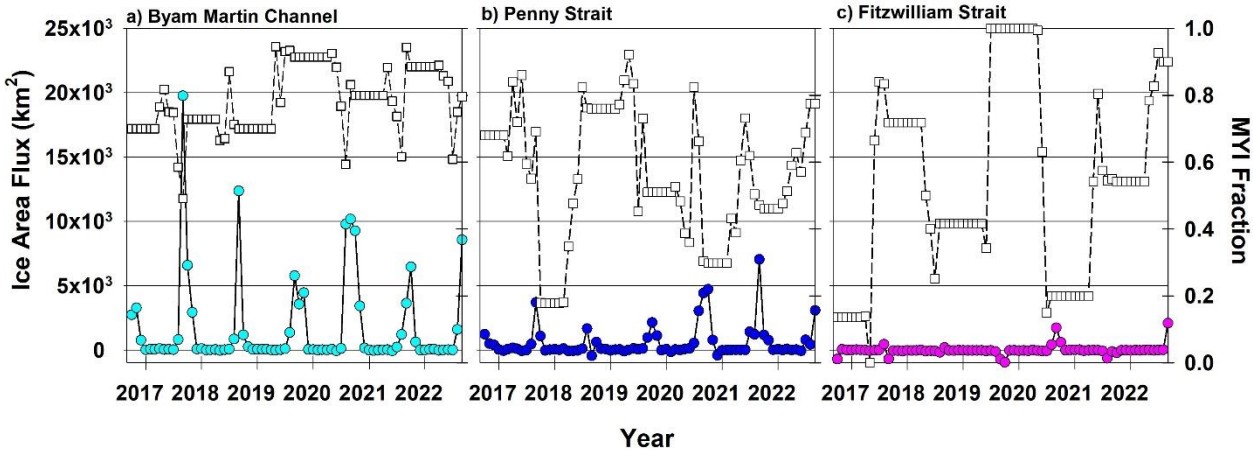

**Figure 7:** Time series of the monthly ice area flux for (a) Byam Marin Channel, (b) Penny Strait, and (c) Fitzwilliam Strait October 2016 to September 2022. Square boxes indicate multi-year ice (MYI) fraction. Positive flux values indicate ice export from the Queen Elizabeth Islands to the Parry Channel and negative flux values indicate ice import from the Parry Channel to the Queen Elizabeth Islands.

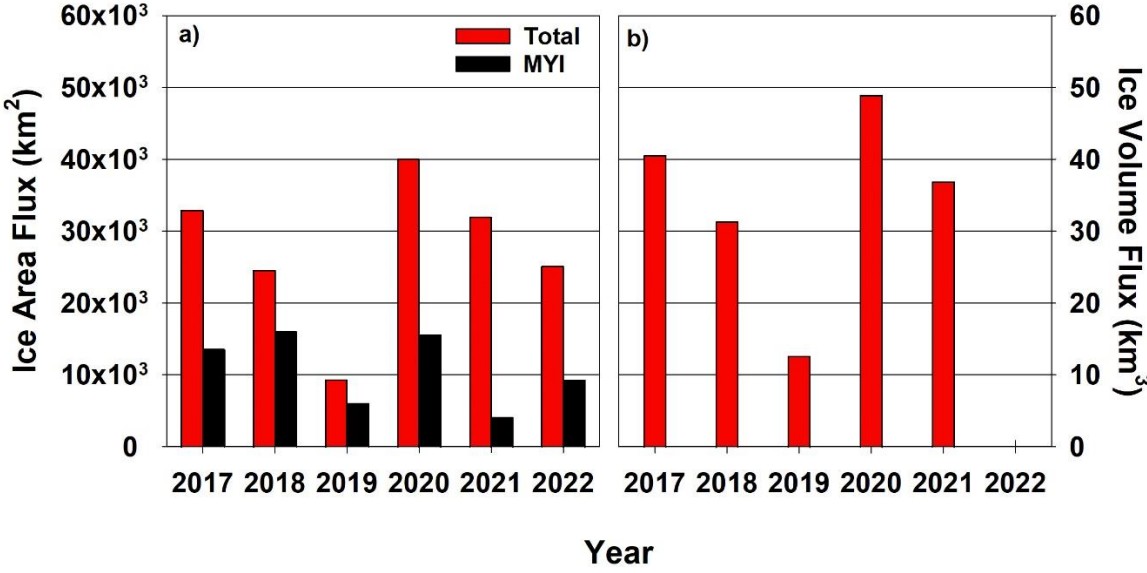

**Figure 8:** Annual (October to September) (a) total and multi-year ice (MYI) ice area flux and (b) ice volume flux from the Queen Elizabeth Islands to the Parry Channel for 2017 to 2022. No annual ice volume data was available for 2022.




## 4.4 MYI replenishment in the CAA

The two components that replenish the CAA's inventory of MYI are i) MYI that is imported from the Arctic Ocean (dynamic) and ii) FYI that survives the melt season and is promoted to MYI (thermodynamic). Howell et al. (2015) quantified these MYI replenishment components from 1997-2013 using the same ice area flux method used in this study for the dynamic component and the Canadian Ice Service ice charts for the thermodynamic component. Indeed, Canadian Ice Service ice charts can be used to estimate FYI aging by taking the sum of all SYI within the CAA on the first weekly CIS ice chart of October. The first week of October is used because this is when all remaining FYI within the CAA that survived the melt season gets promoted to SYI.

Howell et al. (2015) reported that the total MYI replenishment from 1997-2013 was an average of $65x10^3$ km$^2$ with $13x10^3$ km$^2$ representing dynamic import and $52x10^3$ km$^2$ representing thermodynamic survival. However, the dynamic component estimate was limited to May to November due to SAR image availability. Since our sea ice area flux estimates in this study are year-round it provides an opportunity for more robust estimates of MYI replenishment within the CAA. Figure 9 shows the annual MYI replenishment within the CAA for our 6-year study period. The 6-year annual average for MYI replenishment was $72\pm59x10^3$ km$^2$ with the dynamic component being $16\pm49x10^3$ km$^2$ (22%) and thermodynamic component being $56\pm36x10^3$ km$^2$ (78%). These values are relatively similar to the replenishment components from 1997-2013 reported Howell et al. (2015) and suggest the processes of MYI replenishment within the CAA have not changed appreciably over the last 25 years. This is a considerable contrast to the abrupt disruptions in MYI replenishment processes operating in the Arctic Ocean (Babb et al., 2023). It also reinforces the fact that MYI replenishment within the CAA is primarily the result of FYI surviving through summer rather than the annual import of MYI from the Arctic Ocean into the QEI. However, given that Arctic Ocean MYI import is increasing (Howell and Brady, 2019) together with thinning landfast ice in the CAA (Howell et al., 2016; Glissennaar et al., 2023) becoming more prone to melt it seems likely that the relative contribution of these two sources of MYI will change in the future.

Although over longer-term periods the MYI replenishment mechanisms appear to be stable, on an interannual basis the two can vary significantly. For instance, strong positive net MYI replenishment was mainly caused by FYI survival in 2017, 2018, and 2021 (Figure 9). In 2021 the area of ice surviving summer melt was much lower, but this was offset by strong MYI import into the QEI and also led to a year of high net replenishment. In contrast, there was net MYI export from the CAA in 2019 and 2022 but this was not offset by strong thermodynamic survival, so the net replenishment in 2019 was >5 times smaller than the other years, and in 2022 there was actually a net loss of $-25x10^3$ km$^2$ MYI.





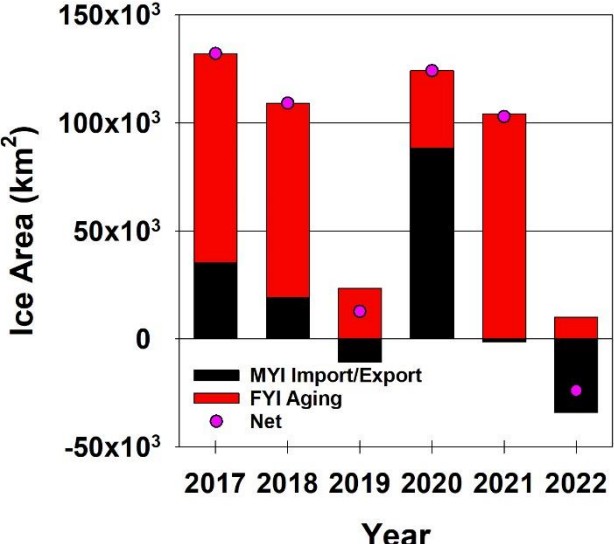

**Figure 9: Annual (October to September) multi-year ice (MYI) replenishment within the Canadian Arctic Archipelago (CAA) for**
**2017 to 2022. Positive MYI values indicate ice import from the Arctic Ocean or Baffin Bay to the CAA and negative values**
**indicate MYI export from the CAA.**

## 5 Conclusions

We summarized the sea ice area transport across the CAA from October 2016 to September 2022 and sea ice volume
transport across the CAA from October 2016 to September 2020. Over the time periods, the annual ice area flux average was
$-134\pm72\times10^3$ km$^2$ and the average volume flux was $-40\pm74$ km$^3$ indicating that the CAA exported more ice to the Arctic
Ocean and Baffin Bay than it received. We demonstrated that the majority of the CAA's ice area and volume flux is exported
to Baffin Bay while the QEI is the only area where ice is routinely imported. On an annual basis, the CAA always acted as a
conduit for ice area but appreciable import of thicker ice from the Arctic Ocean via the QEI can transform the CAA into a
sink for ice volume. The latter was the case in 2020 and almost the case in 2017.


Compared to Nares Strait, the ice area flux from the entire CAA is considerably larger than the ice volume flux on an annual
basis with relative contributions of 141% and 23%, respectively. As a result, the CAA is a larger contributor for ice area but
not for ice volume with respect to the transport of freshwater (as solid ice) to the North Atlantic compared to Nares Strait.
This also emphasizes the importance of thick ice north of the CAA and Greenland that exits through Nares Strait in
impacting ice volume transport downstream to the North Atlantic. Fram Strait still provides considerably more downstream
ice transport to the North Atlantic than both Nares Strait and the CAA combined.

We also provided the first estimates of the sea ice area and volume flux within the CAA through the major passageways of the QEI to the Parry Channel. Most of the ice leaving the QEI was via Byam Martin Channel followed by Penny Strait and it was negligible at Fitzwilliam Strait. This confirms previous suggestions by Melling (2002) and Howell et al. (2009) that just

south of Byam Martin Channel is the key "choke point" for marine navigation through the Northwest Passage. The ice flux from the QEI to the Parry Channel primarily occurred in summer months but was substantial with annual averages of $27\pm10\text{x}10^3$ km$^2$ and $41\pm15$ km$^3$ for ice area and volume flux, respectively.

Although the time series of this study was insufficient to examine long-term trends, Arctic Ocean ice import into the QEI has

increased since 1997 (Howell and Brady, 2019) which means that it is unlikely the ice flux from the QEI southward to the Parry Channel has decreased. As long as dynamic processes continue to create thick ice along the north facing coast of the CAA, the transport of thick sea ice from the Arctic Ocean southward through the CAA will continue. This southward transport of ice presents a considerable risk to safely navigating the Northwest Passage, a risk that seems unlikely to drop in the near future.


Finally, we provided more robust estimates of MYI replenishment within the CAA on an annual basis. The 6-year annual average of MYI replenishment was $72\pm59\text{x}10^3$ km$^2$ with the dynamic ice import component contributing $16\pm49\text{x}10^3$ km$^2$ and the thermodynamic FYI survival component contributing $56\pm36\text{x}10^3$ km$^2$. MYI replenishment within the CAA from 2016 to 2022 was similar to estimates from 1997-2013 by Howell et al. (2015). This suggests that despite climate warming

diminishing the sea ice cover in all seasons, across the Arctic, the processes that contribute to MYI replenishment within the CAA continue to operate and appear to have yet been severely impacted.

*Acknowledgments.* D. Babb is supported by the Canada Excellence Research Chair in Arctic Ice, Freshwater Marine Coupling and Climate Change held by D. Dahl-Jensen at the University of Manitoba. J. Landy acknowledges support from the INTERAAC project under grant

#328957 from the Research Council of Norway (RCN), from the Fram Centre program for Sustainable Development of the Arctic Ocean (SUDARCO) under grant #2551323, and from the European Research Council under grant SI3D/101077496.

*Data Availability.* The monthly sea ice area and volume flux data will be provided via the Environment and Climate Change Canada's Open Data Catalogue upon paper acceptance.


*Author contributions.* SELH wrote the manuscript with input from DGB, JCL, IAG, KM, BM, and MB. SELH and MB preformed the analysis. JCL and IAG provided the ice thickness data.

*Competing interests.* SEL is a member of The Cryosphere Editorial board.



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
