# Peer review of "Sea ice transport and replenishment across and within the Canadian Arctic Archipelago: 2016-2022"

_EGUsphere, 2023_

## Author Comment (AC1)

Review of "Sea ice transport and replenishment across and within the Canadian Arctic Archipelago: 2016-2022" by Howell et al.

Summary:
This study focuses on quantifying sea ice transport and replenishment across and within the Canadian Arctic Archipelago (CAA), and particularly along the critical segment of the Northwest Passage shipping route, spanning from the Queen Elizabeth Islands to the Parry Channel. Results indicate that the CAA functions as both a source and sink for sea ice, exporting significant amounts to the Arctic Ocean and Baffin Bay. The study underlines the resilience of multi-year ice (MYI) replenishment within the CAA, with ongoing import from the Arctic Ocean and retention of first-year ice (FYI). The authors emphasize the persistent risk that sea ice poses to key shipping routes in the CAA, including the Northwest Passage, due to substantial ice flux and sustained MYI replenishment.

This study makes use of high-resolution drift data from SAR as well as CryoSat-2 altimetry and can serve as a baseline study of sea ice fluxes in the CAA. While the method itself (sea ice flux estimation) is not new, the study region, especially inside the CAA is rather understudied compared to Fram Strait for example.

General Comments:
To my knowledge there is no comparable study for ice fluxes within the CAA currently available. Therefore, I think it potentially deserves publication. I also had no problems to follow the text in general. The applied methods seem generally solid, but there are some details and decisions that I find disputable and that need clarification and more in-depth descriptions and discussion. Here and there, the paper lacks explanations. I also find the figures could partly contain more information. Major comments are:

Howell et al.
We thank the Reviewer for their comments and have tried to implement all the suggestions.

1. More information on the input data is needed. The authors use two sea ice thickness data sets to compute volume fluxes. The all-year CryoSat-2 summer sea ice thickness retrieval from Landy et al. (for the outer gates) and the proxy-record from Glissenaar et al. (for the inner gates of the CAA). I think it should be discussed how consistent these data sets are. How do they compare at intersections? Because any inconsistency might introduce biases here. I also suggest introducing acronyms or at least make it clearer when and where which data set is used (See also in the specific comments). It is also not entirely clear over which period data have been used. Under "2) Data", it is stated that volume fluxes have been calculated until 2021, but later in Figure 4, it is only until 2020, while the caption mentions until 2022. This is confusing. It should be possible to provide volume fluxes until 2022, I assume.

Howell et al.
We have been able to update the outer gates temporal domain to match the inner gate temporal domain for volume flux. The time series of sea ice thickness data is October 2016 to September 2021 for both gates therefore, the area flux is 6-years and volume flux is 5-years. We make this

clear in the text. The proxy-record from Glissenaar et al. for the inner gates was actually generated from the winter observations in the Landy et al. CryoSat-2 product. The CryoSat-2 observations were used as training data for the machine learning algorithm in Glissenaar et al. Thus, when the ice thickness products are compared against each other there is understandably no bias, but the RMSE is 41 cm (Glissenaar et al. 2023) which was also added to the text.  We refrain from acronyms which we feel they make papers more confusing (i.e. spelling it out is better especially for casual readers). We have revised the dataset section as follows:

Sea ice thickness estimates for the outer flux gates were acquired from the CryoSat-2 radar altimeter from October 2016 to September 2021 (Landy et al., 2022) that uses a combination of data from Landy et al., (2020) for the 'cold' season (October to April) and Dawson et al., (2022) for the summer period (May to September). A bias correction based on radar model simulations (Landy et al., 2022) is applied to the summer radar freeboards. The entire time series of radar freeboards is then converted to a continuous pan-Arctic record of sea ice thickness using snow loading information from SnowModel-LG (Stroeve et al., 2020) and assuming constant ice-type dependent densities for sea ice (Landy et al., 2022). All CryoSat-2 ice thickness data are available from https://data.bas.ac.uk/full-record.php?id=GB/NERC/BAS/PDC/01613. No outer gate sea ice thickness estimates were available from October 2021 to September 2022 (i.e. the 2021/2022 ice season).

Sea ice thickness estimates for the inner flux gates within the CAA were obtained from the ice thickness proxy record developed by Glissenaar et al. (2023), which is available from https://doi.org/10.5281/zenodo.7644084.  This proxy sea ice thickness dataset uses the CryoSat-2 observations of sea ice thickness from Landy et al. (2022) in the open seas of the Canadian Arctic to train a random forest regression model to estimate sea ice thickness from information in the Canadian Ice Service ice charts (Tivy et al., 2011) within the channels of the CAA. The uncertainty of the proxy sea ice thickness values ranges from 30 to 50 cm. When the proxy dataset is compared against independent CryoSat-2 thickness values, not used for training, the root mean square error (RMSE) is 41 cm (Glissenaar et al. 2023). Unlike direct CryoSat-2 observations, this proxy sea ice thickness dataset is available in all channels in the CAA from 2016 to 2021 but only for the months of November to April. No inner gate sea ice thickness estimates were available from October 2021 to September 2022 (i.e. the 2021/2022 ice season).

2. I find the method description of how area and volume fluxes are computed lacks information. How are sea ice motion, concentration, and thickness co-registered along the gate? Are you using a nearest-neighbor scheme? How are data gaps handled? Are you filling those by interpolation? I also recommend adding a figure showing the drift, concentration, and thickness along a few gates, to get an impression of spatial resolution of input data and how thickness is distributed along the gates.

Howell et al.
Admittedly, we were too vague on the methods because they have been described in previous papers going back to 2013.  We have added a Figure (Figure 2) to show the flux across the gate but showing thickness is not really helpful as it is just an average within the buffer. We have updated the methods as follows:

Our approach to estimate the sea ice area flux from sequential pairs of SAR imagery is robust and based on previous work (e.g. Kwok, 2006; Agnew et al., 2008; Howell et al., 2013). For each SAR image pair, sea ice motion was estimated using the Environment and Climate Change Canada Automated Sea Ice Tracking System (ECCC-ASITS; Howell et al., 2022) that is based on the Komarov and Barber (2013) feature tracking algorithm. Sea ice motion estimates are then interpolated to a 30 km buffer region around each gate using inverse distance weighting. Both sea ice motion and Canadian Ice Service ice concentration values are then sampled at 5 km intervals along the gate as shown in Figure 2.

…

The sea ice volume flux of the CAA's outer gates from October 2016 to September 2021 was determined from the product of the monthly ice area flux and the monthly average CryoSat-2 sea ice thickness within the 200 km buffer around each gate. For the inner gates the volume flux was determined from the product of monthly ice area flux and the monthly average proxy sea ice thickness. Note we use the larger (i.e. 200 km) buffer for volume flux given the coarse spatial resolution of the thickness products (i.e. 50-80 km).

[Figure]

Figure 2. Sea ice motion vectors overlayed with sea ice concentration from the Canadian Ice Service ice charts for the Lancaster Sound gate. The yellow region represents the 30 km buffer zone around the gate. RADARSAT Constellation Mission (RCM) imagery on April 8, 2022 (RCM © Government of Canada).

3. I find the approach of using a linear trend to bridge the summer gap for the thickness at the inner gates is daring. Are there in situ measurements like from buoys or other observations that support this approach? At least uncertainties should be significantly higher.

Howell et al.
We agree it is "daring" but there are no buoys or in situ measurements to compare against. In fact, there is an overall lack of sea ice thickness measurements in the CAA except for a few winter survey's (Melling, 2002; Haas and Howell, 2015), landfast ice locations (Howell et al., 2016). In the summer months there is pretty much nothing. Therefore, we would like to keep the volume flux estimate in the paper because it would be considered a baseline estimate until it can be refined. Linear interpolated thickness uncertainty is unquantifiable in a meaningful way therefore, we provide a Figure (Figure 3) to indicate and state that the volume flux estimates are likely overestimated over the annual cycle and have revised the text as follows:

Since sea ice thickness values from the proxy ice thickness dataset were not available from May to October, accordingly we used the linear trend of April to November to approximate those values. This follows the climatological record of landfast ice thickness loss through melt in summer that was reported for Eureka by Dumas et al. (2006) in their Figure 9. The reduction in ice thickness at Eureka follows an approximate linear trend between June and September. Figure 3 here shows a time series of plot of CryoSat-2 sea ice thickness at the QEI gates (in black) followed by the time series proxy ice sea ice thickness at Byam Martin Channel (in red), and the linear interpolated ice thickness at Byam Martin Channel (dashed red). Note the thickness decrease with latitude is similar to what has been reported in previous studies (Melling, 2002; Haas and Howell, 2015). With the exception of 2016-2017, the linear thickness approximation likely overestimates thickness for 1-3 months per year (Figure 3) and as a result the inner gate annual volume flux estimates are likely to be overestimated.

[Figure]

Figure 3. Time series of CryoSat-2 ice thickness for the Queen Elizabeth Islands (QEI; black) outer gates, proxy ice thickness for the Byam Martin Channel (BMC, red) inner gate, and linearly interpolated ice thickness for BMC inner gate (dashed red) from 2016-2022.

4. The authors calculate uncertainties for both area and volume fluxes, and provide estimates in a table, but it is quite difficult to relate them to the flux estimates in the figure. I strongly recommend adding error bars in the figures where you provide flux estimates.

Howell et al.
Agreed. We have placed error bars on the flux Figures.

5. There is the study of Agnew et al. (2008) that provides area fluxes across the CAA. Can you compare area fluxes with those of Agnew et al.?

Howell et al.
That comparison, in addition to ice area fluxes from Kwok (2006) was done in a previous study for the Arctic Ocean outer gates (Howell et al., 2013). We could compare with Lancaster Sound and Jones Sound but we are not really looking at those gates at the monthly scale and Agnew et al. (2008) did not have sufficient data for annual estimates (i.e. losing the ice surface at 89 GHz). The comparison for the Arctic Ocean facing gates was good and the approach is identical so they should be comparable.

More specific comments:
L96: I thought the Landy et al. data set already combines summer and winter sea ice thickness from CryoSat-2: "A year-round satellite sea-ice thickness record from CryoSat-2"? Did you use these data? In the text it sounds like you did the combination of summer and winter data yourself.

Howell et al.
Landy et al. (2022) does do that as it combines Landy et al. (2020) and Dawson et al. (2022). We clarified this in the text as follows:

Sea ice thickness estimates for the outer flux gates were acquired from the CryoSat-2 radar altimeter from October 2016 to September 2021 (Landy et al., 2022) that uses a combination of data from Landy et al., (2020) for the 'cold' season (October to April) and Dawson et al., (2022) for the summer period (May to September). A bias correction based on radar model simulations (Landy et al., 2022) is applied to the summer radar freeboards. The entire time series of radar freeboards is then converted to a continuous pan-Arctic record of sea ice thickness using snow loading information from SnowModel-LG (Stroeve et al., 2020) and assuming constant ice-type dependent densities for sea ice (Landy et al., 2022).

L118: The information on the apertures can be provided either in a table or better in Fig. 1, avoiding listing it in the text.

Howell et al.
We do not feel a table is needed for this basic information.

L154: "was determined from the product of the monthly ice area flux and the monthly average CryoSat-2 sea ice thickness" – In the method section you write that you use different thickness data sets (The Landy record and the proxy record) for the inner and outer gates. Please clarify.

Howell et al.
Agreed we clarified as follows:

The sea ice volume flux of the CAA's outer gates from October 2016 to September 2021 was determined from the product of the monthly ice area flux and the monthly average CryoSat-2 sea ice thickness within the 200 km buffer around each gate. For the inner gates the volume flux was determined from the product of monthly ice area flux and the monthly average proxy sea ice thickness. Note we use the larger (i.e. 200 km) buffer for volume flux given the coarse spatial resolution of the thickness products (i.e. 50 km for inner gates and 80 km for outer gates).

L155: This is because you use the "proxy record" here, right? Perhaps mention that. However, I wonder how robust it is to just interpolate between April and October. How do you estimate the volume flux uncertainties in the CAA in summer? Are there any in-situ data to compare with?

Howell et al.
We addressed this in response to Major Comment 3 above.

Figure 2: I suggest adding error bars with the calculated uncertainties. Moreover, in the caption and/or the figure itself, it should be written more clearly which flux and gates/regions are considered here.

Howell et al.
Agreed.

Figure 3: I suggest to also add error bars here. Moreover, in the caption, I assume it should be "NET export/import"?

Howell et al.
Agreed.

Figure 4: Same comment as for Fig. 3. Moreover, the caption says "for 2017 to 2022" – but only 2017-2020 is shown?

Howell et al.
Changed.

---

## Author Comment (AC2)

The study explores ice area and volume fluxes into, around, and out of the Canadian Arctic Archipelago (CAA) using a combination of remote sensing datasets. Ice motion data are derived from SAR imagery for deriving area fluxes and thickness estimates from CryoSat-2 around the primary gates and an ML random forest model for the inner gates. The paper follows on from a study by the same author in 2022 deriving the new ice area flux estimates from Sentinel-1 and RADARSAT RCM (Howell et al., 2022), a paper in 2022 looking at MYI conditions in the CAA (Howell et al., 2023a), then a paper in 2023 looking at similar ice area/volume fluxes of the CAA and Nares Strait by integrating the new CryoSat-2 thickness data with the previous area flux estimates (Howell et al., 2023b).

Overall, the analysis in the paper was well presented and relatively easy to follow and the science aligns well the scope of The Cryosphere. My comments are provided below, I hope these make sense and help improve the manuscript.

Alek

Howell et al.
We thank Dr. Petty for the review comments and have tried to implement all of the suggestions.

General points:
1. The first thing I think worth flagging is how closely aligned this paper is with the 2023 JGR study (Howell et al., 2023) that derived similar estimates of ice area and volume fluxes using the same method and over the same time-period but also comparing to Nares Strait. The methods sections of both papers are virtually the same, which I think is fine, but I was surprised there wasn't a clearer note to this effect in this paper. The addition in this study seems to be the CAA inner gate area/volume flux estimates that were not included in the 2023 study and the generally increased focus on the CAA results. Still, it took me a bit of a while to realize that and I think this paper should tie together much more with that study and make clearer what the relative goals of each study are and what exactly is new in this study. Similarly, the MYI replenishment section didn't refer to the author's recent paper on CAA MYI replenishment which seemed a little odd too. Put another way, the author and author team have been looking closely at area/volume/MYI fluxes around the CAA for a while now, so what was the gap that this study needed to respond to and how did it build on the preceeding efforts?

Howell et al.
The previous paper (Howell et al., 2023b) only focused on Arctic Ocean export (i.e. inflow into the Canadian Arctic domain). This paper is focused on inflow and outflow for the CAA together with internal flux estimates. As for MYI replenishment, it was only looked at in this context in the 2015 paper (Howell et al., 2015). The recent MYI paper used Ice Charts to estimate dynamics, here we can use SAR to get more robust estimates. Altogether, we understand the Reviewer's point and have revised accordingly as follows:

Specifically, previous studies using SAR imagery have only been able to quantify ice flux between Arctic Ocean and CAA (Kwok, 2006; Howell et al., 2013; Howell and Brady, 2019) ignoring ice flux between the CAA and Baffin Bay. This omission also constrained MYI replenishment estimates within the CAA (Howell et al., 2015). Limited SAR image availability has also prevented ice flux estimates from the QEI to the Parry Channel which is a key part of

the Northwest Passage. Using SAR imagery from Sentinel-1 and RCM Howell et al., (2023b) was able to provide year-round flux estimates between the Arctic Ocean and CAA but did not consider the ice flux between the CAA and Baffin Bay.

2. I'm not a fan of the uncertainty bounds approach, especially as they aren't really used (hard to plot an uncertainty bounds in a time-series plot!). I really think you should just pick your best guess uncertainty estimate and justify it as best you can – I don't particularly believe the bounds truly represent realistic bounds anyway. It would be good to then use those values on the time-series plots you show to get a sense of how important the uncertainties are for assessing seasonal/annual variability. However, I do have some additional concerns about the uncertainty quantification:

Howell et al.
Agreed. We selected the upper bound uncertainty and applied it to all Figures.

     1. What about the errors in ice concentration (L158)? I would guess they are not negligible considering how small some of the channels are, but maybe this was addressed in one of the recent papers.

Howell et al.
Ice concentration estimates are from the Canadian Ice Service (CIS) ice charts which do not suffer from the problem of small channels and surface melt ponds like passive microwave (e.g. Agnew and Howell, 2003). Further, since 1995 they are derived entirely from high spatial resolution synthetic aperture radar (SAR) imagery. They are often used as validation (i.e. truth) so one could argue the errors in ice concentration are negligible compared to other uncertainty components of the flux calculations.

     2. Seems quite odd to assume the ice motion errors are uncorrelated if derived based on the same image pair? But then you later add the errors to generate the monthly uncertainty estimates so you assume they are correlated? Are these assumptions justified better in previous papers?

Howell et al.
These are widely used assumptions based on previous papers (i.e. Kwok and Rothrock, 1999; Kwok, 2006; Agnew et al., 2008; Kwok et al., 2010) that we acknowledge. It is indeed an image pair, but each image is different so the uncertainty estimates between spatially adjacent pairs are assumed to be uncorrelated. We are not adding errors but estimating the error per month based on the number of observations (i.e. image pairs) which is approximately 1 per day or ~30 per month, and the daily ice motion uncertainty estimates are assumed to be correlated.

     3. The use of the linear trend to fill in the summer months for the inner gates seems very crude, especially as this seemed to be one of the big differences with the 2023 JGR study. At the very least I think the paper would benefit from showing what the raw and interpolated thickness values actually look like at each gate and how justifiable they are. Any way you could tie it together with the all-season CS-2 data outside the CAA?

Howell et al.

Indeed it is crude but there is an overall lack of sea ice thickness measurements in the CAA except for a few winter survey's (Melling, 2002; Haas and Howell, 2015), landfast ice locations (Howell et al., 2016). In the summer months there is pretty much nothing. Therefore, we would like to keep the volume flux estimate in the paper because it would be considered a baseline estimate until it can be refined. Linear interpolated thickness uncertainty is unquantifiable in a meaningful way therefore, we provide a Figure (Figure 3) to indicate and state that the volume flux estimates are likely overestimated over the annual cycle and have revised the text as follows:

Since sea ice thickness values from the proxy ice thickness dataset were not available from May to October, accordingly we used the linear trend of April to November to approximate those values. This follows the climatological record of landfast ice thickness loss through melt in summer that was reported for Eureka by Dumas et al. (2006) in their Figure 9. The reduction in ice thickness at Eureka follows an approximate linear trend between June and September. Figure 3 here shows a time series of plot of CryoSat-2 sea ice thickness at the QEI gates (in black) followed by the time series proxy ice sea ice thickness at Byam Martin Channel (in red), and the linear interpolated ice thickness at Byam Martin Channel (dashed red). Note the thickness decrease with latitude is similar to what has been reported in previous studies (Melling, 2002; Haas and Howell, 2015). With the exception of 2016-2017, the linear thickness approximation likely overestimates thickness for 1-3 months per year (Figure 3) and as a result the inner gate annual volume flux estimates are likely to be overestimated.

[Figure]

Figure 3. Time series of CryoSat-2 s ice thickness for the Queen Elizabeth Islands (QEI; black) outer gates, proxy ice thickness for the Byam Martin Channel (BMC, red) inner gate, and linearly interpolated ice thickness for BMC inner gate (dashed red) from 2016-2022.

4. It would be good to get a better sense of how important the results are to the thickness estimates, I'm guessing there is some skill in the seasonal thickness cycle

in the input datasets but not much de-seasonal skill beyond that considering the errors.

Howell et al.
Adding the ice thickness data for volume instead of area estimates provides the additional dimension and less uncertainty than the ballpark estimates of e.g. early Kwok paper who lacked thickness estimates. These ballpark estimates currently act as the baseline for SIV fluxes within the CAA.

It is true that the ice thickness estimates are more skillful at measuring the average seasonal cycle than measuring interannual variability. However, in both papers describing the ice thickness datasets used here, Landy et al. 2022) and Glissenaar et al. (2023), the authors provided anomaly correlation coefficients of sea ice thickness time series versus in situ draft/thickness estimates obtained from mooring-based ULS or landfast ice drill hole sites. For both products, the ACCs were lower than the correlations for seasonal time series, but all positive and between 0.11 and 0.51, meaning that oscillations between higher- and lower-than-normal thickness are always captured and for some locations the magnitude too with significant skill. Our SIV fluxes will integrate any skill capturing the IAV, which should be an improvement compared to using ballpark or climatological mean thickness data. This point has now been emphasized in the manuscript.

3. L266-266 on QEI area vs volume import/export I view as the most interesting idea from the paper but think it should be explored in much more detail to help justify this paper. How thick does the ice need to be north of QEI for the assumptions of net volume sink to be true? Do we think we're approaching an inflection point of this not being true anymore? I think it would be easy and quite illuminating to run a little sensitive test here changing the ice thickness values north of QEI and re-running the analysis. I don't know the cited Melling 2022 study that well but the claim that the thickness isn't changing north of QEI is a little surprising to me, but eventually I think we can agree it's quite likely to change after a big MYI flushing event. In general, I think the paper would be much improved if you could test some hypotheses out in this framework rather than just showing raw data and discussing ideas.

Howell et al.
We appreciate the Reviewers suggestion and agree import from the QEI has the most impact on the volume balance of the CAA. However, it is not really how thick the ice is but rather how much ice comes in. This is controlled by the timing and collapse of ice arches together with winds. The ice north of the QEI is generally thick experiencing only very slight (not significant decline) since? 2016 (see Figure 3 above). So again, ice duration and wind are the key drivers not thickness during this time period. We added this following to the discussion:

Overall, the CAA appears to be a strong ice area exporter and a lower ice volume exporter when appreciable ice from the Arctic Ocean is imported to the QEI. Ice thickness in the vicinity of the QEI has only experienced a slight (not significant) decrease over our study time period (Figure 3). Therefore, the CAA's net volume import in certain years not likely a result of how thick the

ice is but rather how much thick ice is imported into the QEI which is a function of ice arch duration and atmospheric circulation patterns (i.e. wind).

4. Finally, on a similar theme, it's quite hard as a reader to intake all the different flux estimates and get a sense of what it all means. Most of the figures don't really much of a compelling story or scientific result. The discussion and conclusion sections do help but they are not very visual. A map schematic showing the area and volume flux estimates for your study period I think could help a lot?

Howell et al.
Interesting suggestion but with such inter-annual variability this is difficult to depict on a schematic. We did however place arrows on Figure 1 (below0 indicating the dominate direction of transport which should help readers interpretation throughout the paper.

[Figure]

Specific comments:
L33 – are goods actually transported through the NWP?!'

Howell et al.
Yes. Many communities in the Canadian Arctic receive their supplies via ship.

L60 – but then at L66 you say people have done this also using AMSR-E. So, what resolution do we truly need for this kind of analysis?

Howell et al.

89 GHz enhanced imagery at 2.2 km spatial resolution. But Kwok has done it with 6 km. The problem is resolving the summer sea ice conditions is challenging and lower resolution is bias to slower speeds.

L64 – what exactly do you mean by images not being consistent in space and time?

Howell et al.
Change to spatially and temporal uniform to construct ice flux estimates of over all the gates. For example, image availability from RADARSAT-1 and RADARSAT-2 was typically only available with high temporal resolution in certain regions of the CAA and mostly only during operational months of June to October. As a result, a complete picture of sea ice dynamics of the CAA over the entire annual cycle was not possible.

L68. – The relative benefits/merits of S-1/RCM vs R1/R2 for doing this area flux analysis is still a little unclear to me, was this explained better in the earlier papers?

Howell et al.
We addressed this in a previous comment (General Comment 1). Hopefully the benefits of S1/RCM vs R1/R2 are now clear.

Figure 1 – I think it would be good to highlight Baffin Bay/Beaufort Sea as they are mentioned in the text too, appreciate the figure may need to be zoomed out a little more.
Generally confused by the discussion of Parry Channel but Peary Channel being indicated on the map?!

Howell et al.
Parry Channel is correct. The Parry Channel was named after William Edward Parry who "almost" discovered the Northwest Passage. We went with an inset. See response to previous comment.

L96 - this discussion of the all-season CS-2 data is a little odd, you are just using the data from the 2022 paper right, so shouldn't that be cited first? The rest is background to that study.

Howell et al.
Agreed. Rephrased accordingly.

L106 – what are the open seas of the CAA?
Howell et al.
Rephrased to marginal ice regions.

The CAA thicknesses from the Isolde 2023 paper basically show a seasonal cycle of ~60 cm to 160 cm, errors in the proxy data of 30 cm?

Howell et al.
That is a bulk value. Errors range from 30 to 50 cm.

L28 - Not quite sure what you mean by buffer region, this seems maybe a bit colloquial. Can you be more explicit?

Howell et al.
We have added a Figure to illustrate the "buffer region" as follows:

[Figure]

L143 – what are the units of that? Km/day??
Howell et al.
Displacement error should just be km.

L160 – this is confusing as you don't reference the proxy data which I believe is what you are using for the inner gates here. I think in general it would help to show what these data look like for a given gate as a case study – show the area flux, the thickness then the fluxes for a given season with those applied uncertainties. Would help us visualize the variability in the source terms and how it relates to the variability in the flux terms.

Howell et al.
Agreed.  We added a Figure to help visualize the uncertainty. See previous response.

Figure 2 - How well correlated are the area and volume fluxes? Could also show the thickness/are variability too. As stated in the general points, also unsure why the uncertainties are not shown. I think just take your best guess uncertainty and include that in the figures, would help to visualize how they compare to the variability of the signal.

Howell et al.
Agreed.  Added uncertainty to all Figures.

L260 – Shackleton would be turning in his grave over that comment! But seriously, I would guess there is a good chance there could be thicker ice in the Weddell Sea..?

Howell et al.
Perhaps not as north of the CAA and Greenland is likely thicker.  Nevertheless, we added "Arctic."

L318 onwards - I feel like the MYI replenishment ideas could benefit from knowing how much MYI is lost too? Struggling a bit too put the replenishment numbers in context.

Howell et al.
Agreed but it is very tricky to estimate how much MYI actually melts. Especially quantifying how much MYI is advected into the CAA and melts has too much uncertainty associated with it. As it stands, replenishment is a function of advection from the Arctic Ocean and local FYI aging.

L359 – again I think here is where you could benefit from a better understanding of how sensitive this result is to the underlying thicknesses.

Howell et al.
As previously discussed, it is not so much thickness by how much is allowed to pass through the northern CAA (QEI region) which is controlled by ice arch duration and wind.

Figure 3 and 4 – it would be quite easy and I think much better to read if you combined these, put the volume flux alongside the area flux bars with a twinned y-axis on the other side. Ideally it would be good to see the area and thickness numbers too.

Howell et al.
We prefer to keep them separated.

In several figures you should add the exponent multiple to the label and make the figures consistent in this regard.

Howell et al.
We do not label for volume because of smaller values.  We assume that is point raised by the Reviewer.

---

## Author Response (AR2)

Reviewer# 1
2nd Review of "Sea ice transport and replenishment across and within the Canadian Arctic Archipelago: 2016-2022" by Howell et al.

Thank you very much for the revised version. I think most of the points have been addressed. It is appreciated that you updated the outer gates timeseries for volume flux.

The only point remaining is the interpolation between April and November. You refer to Dumas et al. (2006), but I would disagree with "The reduction in ice thickness at Eureka follows an approximate linear trend between June and September." Where is this stated in the paper or shown in the figure? Actually, it seems that observations only cover the winter season until late spring. Moreover, these are historical data from < 1990. Actually, the observations show even an increase until mid-May. Then they apply a thermodynamic model, which indicates linearity only over may be max. 2 months.

So, I find the method of liner interpolation still not very convincing. I believe applying a simple thermodynamic model would already shed more light on the process between April and November. I understand if you thought this is not within the scope of this paper. But then, if you want to keep the linear interpolation, I think there should be more discussion on the biases due to the linear interpolation. For example, how much (quantify) do you expect over/underestimation of ice thickness and volume fluxes in certain months?

I will indicate minor revisions, but I think this last point should be addressed before publication.

Howell et al.
Fair point. However, running a thermodynamic model in this region is not straightforward given the mix of ice types with the CAA (i.e. 50% MYI on average) and indeed not within the scope of this study. The Dumas study was meant to show the linear sea ice (landfast) decline with the CAA but admittedly is not representative within these regions. In these regions, the ice is not stationary and is continuously flowing out of the QEI being subsequently replaced by thicker ice from the north (i.e. less seasonal ablation) over the duration of the melt season. As a result, this would be (is) very challenging to model because it would need to be a coupled model (dynamics and thermodynamics). We also think quantifying a monthly uncertainty is not possible. To that end, we have provided additional text on the potential problem with this method and we feel Figure 3 nicely shows potential under/over estimation in this region.

Revised text in the Methods section:

Since sea ice thickness values from the proxy ice thickness dataset were not available from May to October, accordingly we used the linear trend of April to November to approximate those values. Figure 3 here shows a time series of plot of CryoSat-2 sea ice thickness at the QEI gates (in black) followed by the time series proxy ice sea ice thickness at Byam Martin Channel (in red), and the linear interpolated ice thickness at Byam Martin Channel (dashed red). Note the thickness decrease with latitude is similar to what has been reported in previous studies (Melling, 2002; Haas and Howell, 2015). Looking at Figure 3, it is apparent that the linear thickness approximation could overestimate thickness for 1-3 months during the melt season and

underestimate for the remainder. This variability is influenced by MYI flowing through these gates that is replaced by thicker MYI from the north (i.e. less seasonal ablation) over the duration of the melt season. As a result, the estimated value of uncertainty for the proxy sea ice thickness dataset that ranges from 30 to 50 cm could fluctuate even more during the summer the months thus impacting volume flux estimates in the region. We acknowledge that quantifying this exact amount for the summer months is challenging and therefore there could be more variability in our annual volume flux values from the QEI to the Parry Channel.

Results Section:
It should be noted that there is more uncertainty in the inner gate volume flux estimates compared to the outer gates because we use linearly interpolated ice thickness values during the summer months and as a result, the uncertainty range shown in Figure 10b could be more variable.

Reviewer #2
Thanks to the authors for addressing the concerns from my previous review. Happy to see this published but with a few minor/technical points below I think could be addressed first (I don't need to see this again!).

I'm pleased you changed the uncertainty approach but it seems the text doesn't exactly match how this is reported in the tables now (e.g. you say it's calculated from a. range in sigma u)? It looks like you followed the recommendation to just report a single value for each gate that represents the monthly mean uncertainty for each gate, but you use the upper bounds of the input error terms, right?

Table 1 and 2 captions also need fixing. I also think the values are also stated to too high a precision (1 s.d. probably sufficient) considering these now represent average uncertainties for the gates.

Howell et al.
We cleaned the text mix-up associated with the uncertainty bounds. The precision conforms to a level previous studies report and is fine.

Reviewer #2
The new reference to the Dumas (2006) paper doesn't make sense to me. What exactly are you trying to say here? I looked at that paper and figure which didn't help!

Howell et al.
Agreed. Removed.

Reviewer #2
I still think the description of the CryoSat-2 data is confusing. I think just state you use the all-season dataset from the Landy et al., (2022) paper. Then you can provide a line or two describing the origins of this datasets.

Howell et al.
Agreed.  We just made it simple as follows:

Year-round sea ice thickness estimates for the outer flux gates were acquired from the CryoSat-2
radar altimeter from October 2016 to September 2021 (Landy et al., 2022). All CryoSat-2 ice
thickness data are available from https://data.bas.ac.uk/full-
record.php?id=GB/NERC/BAS/PDC/01613.

Reviewer #2
A few types with the new additions that need to be fixed.

Howell et al.
Cleaned them up.